



# Aircraft-based mass balance estimate of methane emissions from offshore gas facilities in the Southern North Sea

Magdalena Pühl[1], Anke Roiger[1], Alina Fiehn[1], Alan M. Gorchov Negron[2], Eric A. Kort[2], Stefan Schwietzke[3], Ignacio Pisso[4], Amy Foulds[5], James Lee[6], James L. France[3,7,8], Anna E. Jones[7], Dave Lowry[8], Rebecca E. Fisher[8], Langwen Huang[5], Jacob Shaw[5], Prudence Bateson[5], Stephen Andrews[6], Stuart Young[6], Pamela Dominutti[9], Tom Lachlan-Cope[7], Alexandra Weiss[7], Grant Allen[5]

[1]Deutsches Zentrum für Luft- und Raumfahrt (DLR), Institut für Physik der Atmosphäre, Oberpfaffenhofen, Germany
[2] Department of Climate and Space Sciences and Engineering, University of Michigan, Ann Arbor, MI, USA
[3] Environmental Defense Fund, Berlin, Germany
[4] Norwegian Institute for Air Research (NILU), Kjeller, Norway
[5] Department of Earth and Environmental Science, University of Manchester, Manchester, M13 9PL, UK
[6] National Centre for Atmospheric Science, University of York, York, YO10 5DQ, UK
[7] British Antarctic Survey, Natural Environment Research Council, Cambridge CB3 0ET, UK
[8] Department of Earth Sciences, Royal Holloway, University of London, Egham TW20 0EX, UK
[9] Université Grenoble Alpes, UMR 5001, CNRS, IRD, Grenoble, 38000, France

*Correspondence to*: M. Pühl (Magdalena.puehl@dlr.de)

**Abstract.** Atmospheric methane ($CH_4$) concentrations have more than doubled since the beginning of the industrial age, making $CH_4$ the second most important anthropogenic greenhouse gas after carbon dioxide ($CO_2$). The oil and gas sector represent one of the major anthropogenic $CH_4$ emitters as it is estimated to account for 22% of global anthropogenic $CH_4$ emissions. An airborne field campaign was conducted in April-May 2019 to study $CH_4$ emissions from offshore gas facilities in the Southern North Sea with the aim to derive emission estimates using a top-down (measurement-led) approach. We present $CH_4$ fluxes for six UK and five Dutch offshore platforms/platform complexes using the well-established mass balance flux method. We identify specific gas production emissions and emission processes (venting/fugitive or flaring/combustion) using observations of co-emitted ethane ($C_2H_6$) and $CO_2$. We compare our top-down estimated fluxes with a ship-based top-down study in the Dutch sector and with bottom-up estimates from a globally gridded annual inventory, UK national annual point-source inventories, and with operator-based reporting for individual Dutch facilities. In this study, we find that all inventories, except for the operator-based facility-level reporting, underestimate measured emissions, with the largest discrepancy observed with the globally gridded inventory. Individual facility reporting, as available for Dutch sites for the specific survey date, shows better agreement with our measurement-based estimates. For all sampled Dutch installations together, we find that our estimated flux of (122.7 ± 9.7) kg h$^{-1}$ deviates by a factor 0.7 (0.35-12) from reported values (183.1 kg h$^{-1}$). Comparisons with aircraft observations in two other offshore regions (Norwegian Sea and Gulf of Mexico) show that measured, absolute facility-level emission rates agree with the general distribution found in other offshore basins despite different production types (oil, gas) and gas production rates, which vary by two orders of magnitude. Therefore, mitigation is warranted equally across geographies.





# 1 Introduction

Atmospheric $CH_4$ mole fractions have more than doubled since 1750 due to human activity and continue to rise (Saunois et al., 2020). According to the NOAA Global Monitoring Laboratory, globally-averaged atmospheric $CH_4$ is estimated to have experienced the most dramatic annual increase in 2021 since the beginning of the measurements in 1984 (Lan et al., 2022). With a factor 80-83 times stronger global warming potential over a 20-year time horizon compared to $CO_2$, $CH_4$ is the second-most important anthropogenic greenhouse gas after $CO_2$ and contributes 16% to the effective radiative forcing of well-mixed greenhouse gases over 1750-2019 (Forster et al., 2021). Considering its short life time of around a decade, $CH_4$ bears a high potential for mitigation strategies in order to reach the aim of the UNFCCC Paris Agreement to abate climate warming (Nisbet et al., 2019). Recently, the European Union and the UK signed up to the Global Methane Pledge with the aim to cut global $CH_4$ emissions by at least 30% from 2020 levels by 2030 (European Commission, United States of America, 2021).

The oil and gas sector has been estimated to account for 22 (18-27)% of global anthropogenic $CH_4$ emissions (bottom-up 2017; Saunois et al., 2020). Onboard offshore oil and gas platforms, $CH_4$ is emitted during routine operations due to safety and operational reasons (e.g. shutdown or start-up of equipment during production) by either controlled venting or flaring, i.e., the release of gas or burning of gas. In the latter case, $CO_2$ is released simultaneously, with the $CH_4/CO_2$ emission ratio dependent on the flaring efficiency. According to the United Kingdom Continental Shelf (UKCS) Flaring & Venting report (Oil and Gas Authoritgy (OGA), 2020), in 2019 a total of 2600 metric tonnes (t) $CH_4$ was emitted in the Southern North Sea and the minor Irish Sea region, of which 74% comes from venting, 13% from turbines and engines, 10% are fugitive emissions (e.g. from leaky valves or compressors) and 3% flaring. Carbon dioxide emission was 0.8 Mt in the same year, arising mainly from turbines and engines (95%) with minor contribution of flaring (4%) and venting (0.01%). Flaring accounts for 87% and venting 13% of the total $CO_2$ and $CH_4$ emissions from venting and flaring. Flaring emissions consist of 99% $CO_2$ and 1% $CH_4$ and venting emissions of 98% $CH_4$ and 2% $CO_2$. Dutch $CH_4$ emissions from the extraction of crude oil and natural gas on the Netherlands Continental Shelf (Dutch Pollutant Release and Transfer Register, 2019) amount to 6500 t in 2019, of which 98% comes from venting and fugitives, 1.6% from the usage of natural gas (e.g. as fuel for combustion) and 0.2% from flaring. Carbon dioxide emission was 1.1 Mt with a share of 99% from usage of natural gas, 0.8% from flaring and 0.2% from venting and fugitives. Flaring accounts for 33% and venting/fugitives 67% of the total $CO_2$ and $CH_4$ emissions from venting/fugitives and flaring. Flaring emissions consist of 99.7% $CO_2$ and 0.3% $CH_4$ and venting/fugitive emissions are 89% $CH_4$ and 11% $CO_2$.

In Europe the UK is the second largest and the Netherlands the third largest natural gas producer after Norway (Eurostat, 2018). Most of the UK offshore dry gas production takes place in the Southern North Sea region, which comprises 81 dry gas fields with 181 installations. In 2019, 492 bcf (billion cubic feet) of dry gas was produced. In comparison, the Dutch offshore gas production was 348 bcf from 181 offshore gas fields located in the Southern North Sea.

Several studies indicate that bottom-up inventories underestimate emissions from the oil and gas industry (MacKay et al., 2021; Saunois et al., 2020; Gorchov Negron et al., 2020; Schwietzke et al., 2016; Pétron et al., 2012). Unintended leaks can



significantly contribute to CH$_4$ emissions (Varon et al., 2019; Pandey et al., 2019; Lee et al., 2018; Zavala-Araiza et al., 2017; Conley et al., 2016; Lyon et al., 2015). Top-down emission estimates from direct measurements close to sources can help to independently validate bottom-up estimates in inventory data. A better understanding, monitoring and verification of CH$_4$

70  emissions associated with oil/gas operations is crucial part of the European Methane strategy (European Commission, 2020).

Studies on measurements of CH$_4$ emissions from offshore platforms are still rare. Ship-based measurements were conducted in the U.S. Gulf of Mexico (Yacovitch et al., 2020), in Southeast Asia (Nara et al., 2014) and in the North Sea (Riddick et al., 2019; Hensen et al., 2019). CH$_4$ emissions from the vicinity of 3 UK gas platforms in the Southern North Sea measured by Riddick et al., 2018, are 17.6 - 20.5 kg h$^{-1}$. In this study, observations were taken onboard small boats at an altitude of ~2.5 m

75  (above sea level). The measurements relied on a Gaussian plume model to estimate the vertical resolution of a plume, resulting in a total uncertainty of 45%. Hensen et al., 2019, determined CH$_4$ fluxes around 5 Dutch facilities in the Southern North Sea using a combination of measurements taken 35 m above sea level, a Gaussian plume model and a tracer-release experiment. The results range from 10 kg h$^{-1}$ to 194 kg h$^{-1}$.

In contrast to ship-based measurements, the mobility of aircraft allows for sampling of emission plumes both horizontally and

80  vertically, and thus, airborne measurements provide more detailed information on marine boundary layer conditions which are known to be complex. To the best of our knowledge, the only airborne measurements around offshore facilities conducted so far took place in the Sureste Basin, Mexico (Zavala-Araiza et al., 2021), in the U.S. Gulf of Mexico (Gorchov Negron et al., 2020), in the Norwegian Sea (Foulds et al., 2022; Roiger et al., 2015) and in the North Sea (Lee et al., 2018; Cain et al., 2017). Lee et al., 2018, determined CH$_4$ fluxes higher than 4500 kg h$^{-1}$ arising from an uncontrolled CH$_4$ blow out around one

85  installation in the Central North Sea.

Our paper is organized as follows: In Section 2, we briefly introduce the aircraft instrumentation and sampling strategy applied during the field campaign in the Southern North Sea. We describe the mass balance method used for the calculation of CH$_4$ fluxes and give an overview of the emission inventories. In Section 3, we discuss our measurements and compare the estimated fluxes with the annualized Global Fuel Exploitation Inventory (GFEI) (Scarpelli et al., 2019), UK annually reported data (UK

90  National Atmospheric Emissions Inventory (NAEI), UK Environmental and Emissions Monitoring System database (EEMS)), and with individual reporting by operators of the sampled Dutch platforms. Additionally, we compare our estimated fluxes with ship-based measurements, which were taken around the sampled Dutch platforms in 2018 (Hensen et al., 2019). Finally, we set the findings into a wider context by comparing them with results from aircraft observations in two other offshore regions (Norwegian Sea (Foulds et al., 2022), Gulf of Mexico (Gorchov Negron et al., 2020)).



## 2 Materials and Methods

### 2.1 Campaign 2018/2019 in the Southern North Sea

In April-May 2019 airborne measurements of emissions from offshore installations in the Southern North Sea were conducted within the framework of the United Nations Climate & Clean Air Coalition (UN CCAC) objective to help characterize global $CH_4$ emissions arising from the oil and gas industry. In a previously conducted campaign in 2018, regional survey flights were performed for method development purposes. In 2019, the flight strategy was adapted in order to sample emissions from dedicated installations, which were chosen because of available inventory emission estimates (UK sites, NAEI) and previous ship-borne measurements (NL sites, Hensen et al., 2019). France et al. (2021) describe the instrument payload and the sampling strategy for both campaigns. Here we extend this study with a quantification of $CH_4$ emissions for the studied offshore platforms in 2019.

Figure 1 depicts the flight patterns for 2019. A total of five flights were conducted in the Southern North Sea region. Both UK and Dutch sites of offshore gas facilities were surveyed. One flight (F326) was aborted due to poor weather conditions. Platform positions were taken from the Oil and Gas Authority (OGA) for UK sites and the Dutch Oil and gas portal (NLOG) for Dutch sites. Multiple vertically stacked transects in a 2D plane were flown downwind of targeted platforms to fully capture the vertical extent of a plume. Measurements were made at distances varying from 1 to 10 km from the facilities at altitudes between 45 m - 1300 m above sea level. The flights took place in the afternoon hours, when the boundary layer was expected to be well-mixed. The boundary layer height was determined from abrupt changes in observed potential temperature gradient which mark the boundary layer top, using meteorological data sampled during the vertical profiling of the aircraft.

The DHC6 Twin Otter research aircraft, operated by the British Antarctic Survey (BAS), was equipped with several instruments to collect in situ data of atmospheric trace gases. A Picarro G2311-f 10 Hz Analyser measured dry-air $CH_4$ and $CO_2$ mole fractions at a response time of 0.4 s and at a precision of 1.2 ppb (1σ @ 1 Hz) for $CH_4$. A tuneable infrared laser direct absorption spectrometer (TILDAS, Aerodyne Research Inc.) was deployed to detect $C_2H_6$ (response time < 2 s; precision 50 ppt over 10 s) (Yacovitch et al., 2014). To assess boundary layer physics, sensors for temperature, pressure, humidity and 3D-wind were mounted at the front nose of the aircraft. A NOAA "Best Air Turbulence" probe was installed at the boom of the aircraft and provided wind measurements at a resolution of 50 Hz (Weiss et al., 2011; Garman et al., 2006). More details on the instrumentation and its calibration procedures are given in France, et. al. (2021).





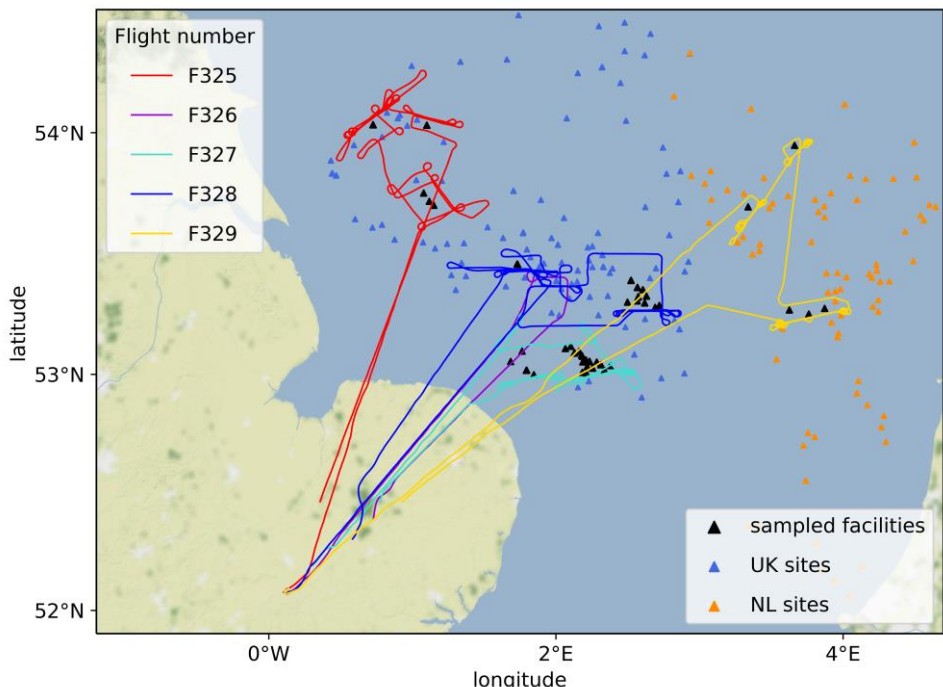

**Figure 1. Aircraft tracks for the 2019 campaign in the Southern North Sea (lines). Location of all offshore facilities in the UK (blue markers) and Dutch (orange markers) region and the sampled facilities (black markers).**

## 2.2 Flux calculation method

We apply the mass balance method to determine the amount of $CH_4$ emitted by the platforms/ multi-platform complexes and passing through a vertical 2D plane downwind (e.g. Pitt et al., 2019; Klausner et al., 2018; O'Shea et al., 2014). For the flux

calculation, measured wind speeds in the target region are required to be relatively steady. Further, under the given meteorological conditions the plume should be vertically well-mixed within the planetary boundary layer. Equation (1) is used to derive the $CH_4$ flux (unit mass per time) across each individual horizontal transect $i$ within the plane, followed by an integration over the vertical plume extent:

$$Flux_i = \Delta C_i \cdot \frac{p_i \cdot M}{R \cdot T_i} \cdot V_\perp \cdot \Delta x_i \cdot D_i \tag{1}$$

$\Delta C_i$ represents the difference of $CH_4$ molar ratios measured in- ($C_i$) and outside ($C_0$) of the plume ($\Delta C_i = C_i - C_0$). $C_0$ thereby denotes the atmospheric background in $CH_4$ and is individually calculated for each transect by a linear interpolation between both plume edges and averaging over a time period of 30 s at either side of the plume. $CH_4$ molar ratios are converted to a $CH_4$ mass density by applying the ideal gas law, i.e. multiplication with molar mass $M$, the ideal gas constant $R$ and measured pressure $p_i$ and temperature $T_i$. The $CH_4$ mass density is then multiplied with the average wind speed $V_\perp$ perpendicular to the

flight track, which is calculated from the measured average wind speed, wind direction and aircraft heading over all transects.



Finally, the CH$_4$ flux for each single transect is obtained by multiplying with the plume width $\Delta x_i$ and the vertical depth of each mixing layer $D_i$. The subdivision of the 2D vertical plane into discrete layers is applied to account for a possible non-uniformly spread (or dispersed) plume. The enhancement measured in each transect is assumed for a layer reaching halfway to the next upper/lower transect. In the case where CH$_4$ enhancements were detected up to the highest transect of the aircraft,

we use the boundary layer height as the maximal upper plume boundary. The boundary layer height is inferred from inspection of the vertical gradient of the potential temperature, which is calculated using the in-situ measured meteorological parameters (Stull, 1988). In case of enhanced CH$_4$ being detected in the lowest transect, the surface is assumed as lower plume boundary. As a result, the bulk net CH$_4$ flux through the plane $Flux_{total}$ is the sum over the fluxes $Flux_i$ calculated for each transect $i$ where CH$_4$ was enhanced:

$$Flux_{total} = \sum_i^{transects} Flux_i \qquad (2)$$

Our flux calculation method is similar to the method applied by Foulds et al. (2022), but differs slightly in the calculation of $\Delta C_i$. Foulds et al. (2022) calculate the background CH$_4$ molar ratio over a greater time period (50 s) due to a more variable CH$_4$ background seen in the Norwegian Continental Shelf. In the Appendix A, the CH$_4$ flux calculation is illustrated by using observations of platform P1 on 30 April 2019. Detailed information on the uncertainty calculation method is provided in the

Appendix B.

**2.3 Emission inventories**

In our comparison with bottom-up estimates we refer to a globally gridded annual inventory based on IPCC Tier 1 methods (IPCC, 2006), UK national point-source inventories and facility-level reporting by Dutch operators.

**2.3.1 Globally gridded annual inventory of CH$_4$ emissions from fossil fuels exploitation**

The Global Fuel Exploitation Inventory (GFEI) (Scarpelli et al., 2019) is a globally gridded 0.1° x 0.1° inventory containing CH$_4$ emissions arising from fossil fuel exploitation for the year 2019. National emission totals, which are based on country-specific emission factors, are reported to the UNFCCC (United Nations Framework Convention on Climate Change) and used in the inventory for a spatial downscaling to the locations of potential sources (Scarpelli, 2020). Thereby, global data sets for oil and gas infrastructure are used. The UK UNFCCC reporting for emissions from the offshore oil and gas exploitation is

based on the UK Environmental and Emissions Monitoring System (EEMS) database (Brown et al., 2022) and the Dutch reporting is based on the Dutch Pollutant Release and Transfer Register (Honig et al., 2022). In the UNFCCC reported data, fugitive emissions are already categorized into subsectors, whereas venting and flaring emissions are reported as totals. Thus, the latter are disaggregated by the inventory to the subsectors using IPCC Tier 1 methods (IPCC, 2006). As a result, the inventory resolves the different fossil fuels sectors (oil, gas, coal) and associated subsectors (distribution (fugitive), exploration

(fugitive + venting + flaring), processing (fugitive, flaring), production (fugitive, flaring), storage (fugitive) and transmission (fugitive, venting)). We compare our emission estimates with the GFEI v2 data set for total global fuel exploitation for gas

from the Harvard Dataverse (Scarpelli et al., 2019). Thereby, we take the inventory data given for each grid cell (Mt/km$^2$) and calculate the emission from the grid cell area.

### 2.3.2 UK annual point-source inventories

The UK Environmental and Emissions Monitoring System (EEMS) database is the environmental database of the UK oil and gas industry maintained by the Offshore Petroleum Regulator for Environment and Decommissioning (OPRED) and the UK Department for Business, Energy & Industrial Strategy (BEIS). It provides annual data from measurements and calculations made for single offshore installations based on reported data from operators. According to the EEMS Atmospheric Emission Calculations (OPRED (BEIS), 2008), monitoring systems of emitted gases are rare at offshore installations. Where no direct
measurement data is available, the emission is calculated by the inventory multiplying activity data (e.g. fuel consumption or flow to flare/venting stack) with locally derived or default emission factors, which are mainly taken from literature. Inventory sources for $CH_4$ and $CO_2$ are differentiated into: engines, heaters and turbines for either diesel, fuel oil or gas consumption; total fugitive emissions; gas flaring from maintenance, routine or upsets/other; total gas venting and emissions from ship oil loading. Latest EEMS data is available for 2018 and 2019.

The UK National Atmospheric Emissions Inventory (NAEI) is an emission database listing all UK point-sources and is provided by BEIS. For offshore oil and gas installations it is based on the EEMS inventory (Tsagatakis, et al., 2022). In the NAEI inventory, emission data is aggregated for all platforms associated with a certain oil or gas field (NAEI, 2020). Offshore emission data is available for $CH_4$ and for $CO_2$. The fluxes observed in this study arise from installations within a certain field and are compared to the inventory data from 2018.

### 2.3.3 Facility-level reporting by platform operators for the survey date

For the sampled Dutch sites facility-level operator-based reporting on $CH_4$ emission was provided after the flights. The OGMP 2.0 level of the reporting corresponds to level 3, i.e. using generic emission factors for individual source types. Additionally, detailed information on operational status, e.g. shutdown, and emission processes (flaring, fugitive, venting) was reported for the specific survey day. Such information was unavailable for the UK facilities upon request via the trade association Oil & 
Gas UK.

### 3 Results

The flight conditions during the flights selected for this study were generally good with moderate wind speeds (< 10 m/s). For one flight the flux calculation for two installations was not successful due to a poorly defined plume. As a result, $CH_4$ emission fluxes have been determined for six UK and five Dutch facilities sampled during flight surveys on 30 April 2019, 2 May 2019
and 6 May 2019, using the mass balance method described above. The installations, for which the flux calculation was

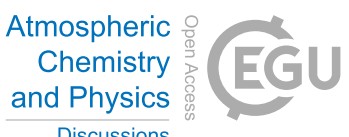

successful, comprise 17% of the UK SNS dry gas production (OGA, 2019) and 6% of the Dutch offshore dry gas production (NLOG, 2019). Under the prevailing conditions found during the three flights, the level of detection, which is a result of the maximum uncertainty of all flux calculation parameters, is 0.3 kg h$^{-1}$ (2$\sigma$). No $CH_4$ enhancement was detected downstream of 4 out of 11 specifically targeted platforms (P3, P5, P6, P9 in Table 1). In addition, a number of several other platforms were passed downwind with no indication of $CH_4$ enhancements. These observations are listed in the Appendix E.

In this section we compare our measured $CH_4$ fluxes with reported emissions and ship-based measurements for Dutch sites. Further, we present observed correlations between $CH_4$, $C_2H_6$ and $CO_2$.

**3.1 Comparison of calculated and reported $CH_4$ fluxes**

In the following, the top-down results of the 2019 measurements are compared to the most recent available bottom-up estimates from globally gridded and national point-source annual inventories from the years 2018 (NAEI, EEMS), 2019 (EEMS, GFEI) and to daily operator-based facility-level reporting. We also compare our results to a ship-based top-down study conducted by Hensen et al., 2019, for the sampled Dutch sites. Observational based top-down methods only provide "snap-shot" emission estimates representing emissions only for the time of the measurements. This means that a) to allow for a comparison the yearly inventory data needs to be scaled to the temporal resolution of the measurement (or vice-versa), and b) a detailed one-by-one comparison is hampered, which is especially true for cases when observations are made during times of non-typical operational conditions, as well as for intermittent emissions (Foulds et al., 2022; Chen et al., 2022). Therefore, for the comparison with inventories a set of "snap-shot" measurements around a group of sites, which represent a distribution of emissions in a region, are preferred over a one-by-one comparison (Tullos et al., 2021).

Figure 2 and Table 1 show the estimated top-down $CH_4$ fluxes along with the reported bottom-up fluxes for all sampled installations P1-P11. Typically, one installation denotes a platform for drilling, accommodation and production. P3 consists of 3 platforms, and P6 has one central platform with 3 satellite platforms. P4 and P5, both multi-platform complexes, have two central platforms with a compression unit and a terminal and several more producing platforms around. P4 consists of two central platforms, 6 platforms for production and 3 wellhead platforms (19 platforms in total). P5 has two central platforms, 4 platforms for production and 3 wellhead platforms (15 platforms in total). Emissions in both regions are the same magnitude and range from 12.1 kg h$^{-1}$ to 86.4 kg h$^{-1}$. Only the multi-platform complex P4 stands out with higher emissions (1258.7 kg h$^{-1}$).





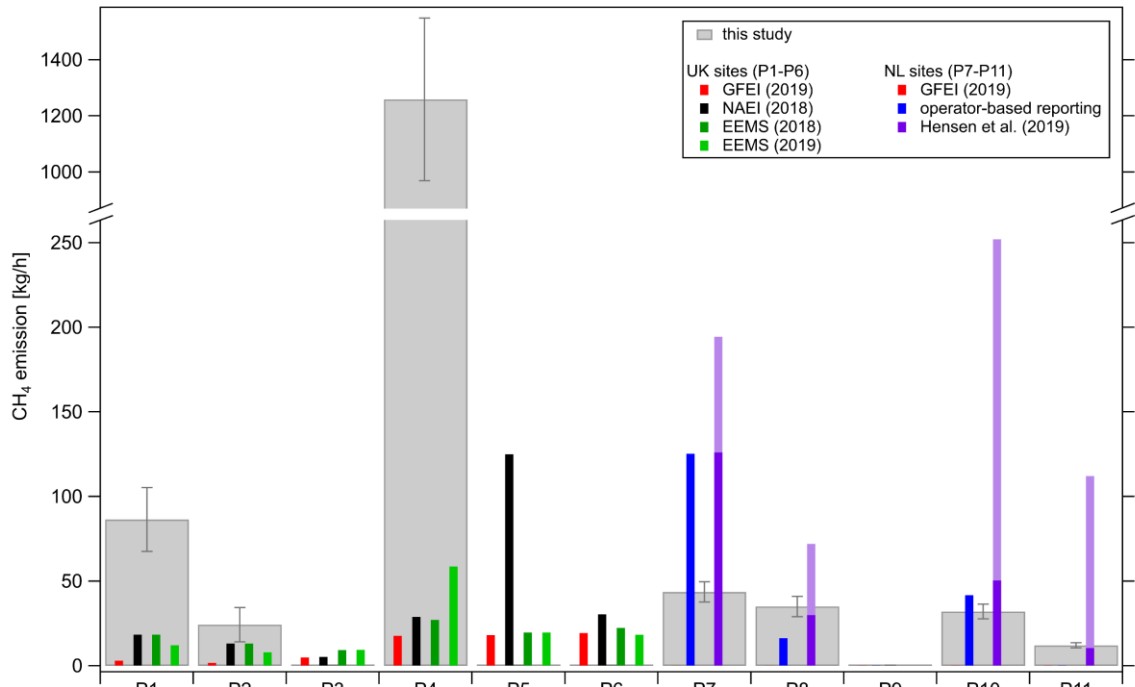

**Figure 2. Comparison of calculated CH₄ fluxes from this study (grey) for UK sites (P1-P6) and Dutch sites (P7-P11) to the Global Fuel Exploitation Inventory (GFEI; red), the UK National Atmospheric Emissions Inventory (NAEI; black), UK Environmental and Emissions Monitoring System database (EEMS; dark green for 2018; light green for 2019), reported fluxes from operators (blue) and a ship-based top-down study (range in light purple; minimal flux in dark purple) for Dutch sites (Hensen et al., 2019). The inventory annual emission data is converted to hourly emissions. For 4 out of 11 targeted installations no downwind enhancements were detected (P3, P5, P6, P9). GFEI (2019) emission data for P7-P11 and operator-based reporting for P9 and P11 is smaller than 0.3 kg h⁻¹. Operator reported values were not available for UK sites.**



**Table 1. Observed CH4 fluxes for UK and Dutch sites from this study and emissions from annual reporting from the UK National Atmospheric Emissions Inventory (NAEI, 2018), the UK Environmental and Emissions Monitoring System database (EEMS, 2018 and 2019) and the Global Fuel Exploitation Inventory (GFEI, 2019). For 4 out of the 11 targeted installations emissions are measured to be below the level of detection (LoD, 0.3 kg h⁻¹). Data from individual operator-based reporting on the specific survey date was available only for Dutch sites. Information on emission processes is given for venting (vent), fugitives (fug) and flaring (flar). Results from a ship-based top-down study (Hensen et al., 2019) is listed for 4 out of 5 sampled Dutch sites.**

| Survey date | 30 April 2019 | | | 2 May 2019 | | | 6 May 2019 | | | | |
|---|---|---|---|---|---|---|---|---|---|---|---|
| Country | UK | | | UK | | | NL | | | | |
| Installation | P1 | P2 | P3 | P4 complex | P5 complex | P6 | P7 | P8 | P9 | P10 | P11 |
| **CH₄ flux [kg h⁻¹]** | | | | | | | | | | | |
| This study — all | 86.4 ± 18.9 | 24.2 ± 10.1 | < LoD | 1258.7 ± 290.5 | < LoD | < LoD | 43.6 ± 6.0 | 35.0 ± 6.0 | < LoD | 32.0 ± 4.4 | 12.1 ± 1.5 |
| GFEI (2019) — all | 2.9 | 1.6 | 4.9 | 18.1 | 19.3 | 19.1 | 0.21 | 0.21 | 0.003 | 0.006 | 0.01 |
| gas processing -fug | 1.3 | 0 | 0 | 0 | 0 | 7.5 | 0 | 0 | 0 | 0 | 0 |
| gas processing - flar | 1.2 | 0 | 0 | 0 | 0 | 7.3 | 0.2 | 0.2 | 0 | 0 | 0 |
| gas production - flar | 0.1 | 0.5 | 1.4 | 5.2 | 5.6 | 1.3 | 0.005 | 0.006 | 0.003 | 0.006 | 0.01 |
| gas exploration - fug, vent, flar | 0.3 | 1.1 | 3.5 | 12.9 | 13.7 | 3.1 | 0 | 0 | 0 | 0 | 0 |
| NAEI (2018) | 18.4 | 13.1 | 5.2 | 28.8 | 124.9 | 30.3 | n.a. | | | | |
| EEMS (2018/2019) — all | 18.4 / 12.1 | 13.1 / 7.9 | 9.2 / 9.3 | 27.1 / 58.6 [a] | 19.6 / 19.6 [b] | 22.3 / 18.3 | n.a. | | | | |
| turbines, engines | 0.4 / 0.4 | 0.5 / 0.4 | 4.7 / 4.7 | 3.8 / 5.5 | 11.4 / 8.2 | 0.01 / 0.7 | | | | | |
| fug | 3.9 / 0 | 6.1 / 0 | n.a. / n.a. | 0 / 0 | 4.5 / 4.6 | 0 | | | | | |
| vent | 14.0 / 11.6 | 6.5 / 7.5 | 4.5 / 4.6 | 23.3 / 53.1 | 3.8 / 6.8 | 22.3 / 17.7 | | | | | |
| flar | 0 / 0 | 0 / 0 | 0 | 0 / 0 | 0 / 0 | 0 | | | | | |
| Reporting by operators (survey date) | n.a. | | | | | | 125.2 vent + fug | 16.2 vent + fug | 0.03 [c] vent + fug | 41.7 vent + fug, no flar | 0 no vent no flar |
| Ship observation (11/2018) (Hensen et al., 2019) | n.a. | | | | | | 126 - 194.4 | 29.9 - 72 | not sampled | 50.4 - 252 | 10.4 - 18.4 |

[a] reporting for 2 (2018) and 1 (2019) platform out of 19 platforms. [b] reporting for 1 (2018 and 2019) platform out of 15 platforms. [c] P9: offline (no dry gas production)



### 3.1.1 Comparison to a globally gridded annual inventory (Tier 1)

We compare our estimated fluxes with the GFEI v2 for 2019, which contains total $CH_4$ emissions from fossil fuels exploitation and a distribution of emissions per subsector. The platforms surveyed in this study are considered to be processing, production and exploration sites by the inventory. As an example, the total $CH_4$ emissions reported for P1 ($2.9$ kg h$^{-1}$) break down to: 44% estimated to arise from fugitives during gas processing, 43% from flaring during gas processing, 10% from exploration (fugitives + venting + flaring emissions) and 4% from flaring during production. According to the inventory, UK emissions

are fugitive, venting and flaring emissions, whereas emissions on Dutch sites arise only from flaring. For all sampled installations, operations other than exploration, production and processing are claimed to emit no $CH_4$.

Compared to the GFEI v2 data set for total $CH_4$ emissions from gas exploitation, the measured fluxes of ($1369.3 \pm 291.3$) kg h$^{-1}$ are 21 times higher than GFEI data ($65.9$ kg h$^{-1}$) for all sampled UK facilities on aggregate. However, the highest emitting UK site (P4 complex) is identified as the highest emitter by the GFEI, as well. The factor by which measured emissions around

Dutch sites are underestimated by the GFEI is an order of magnitude higher compared to UK sites: Measured fluxes (($122.7 \pm 9.7$) kg h$^{-1}$)) are 279 times higher than GFEI data ($0.44$ kg h$^{-1}$) on aggregate for all sites. This high discrepancy points to the weaknesses in using global inventories for field-specific emissions characterisations especially when compared with snap-shot measurement studies. However, similar to UK sites, the two platforms (P7, P8) with highest emissions measured are correctly identified by the GFEI as the highest emitters.

For the sampled installations in this study, Dutch GFEI data is two orders of magnitude smaller compared to UK GFEI data. GFEI relies on UNFCCC reported emissions. Using the UNFCCC GHG Data Interface (UNFCCC, 2022), Dutch annual $CH_4$ fugitive emissions from the natural gas energy production sector and reported for the year 2019 are 14 times smaller compared to the UK equivalent. Further, in contrast to UK reporting, no data is reported for the natural gas subsectors exploration, production and processing. Thus, GFEI values for Dutch sites can only arise from UNFCCC reported total venting and flaring

emissions, since those are disaggregated by the inventory to the subsectors. For the sampled Dutch sites in this study, the inventory gives only flaring emissions from production and processing. Therefore, the UNFCCC reported Dutch emissions, which the inventory is based on, could explain the high discrepancy between GFEI Dutch and UK values.

A related study of 21 oil and gas facilities in the Norwegian Sea finds a better agreement of the GFEI v1 (2016) with the measured fluxes being only a factor 1.4 higher in aggregate for all platforms (Foulds et al., 2022). Similar to the Dutch

UNFCCC reporting, the Norwegian UNFCCC reporting does not show emissions for the natural gas subsectors exploration, production and processing. Considering that Foulds et al. sampled both oil and gas producing installations, the better agreement could possibly be attributed to UNFCCC reported emissions for the oil sector.





### 3.1.2 Comparison to UK annual point-source inventories

The annual estimates of the UK national point-source inventories NAEI and EEMS are smaller than the fluxes measured during
this study. For 2018 the measurement-derived fluxes are a factor of ~6 (NAEI; 220 kg h$^{-1}$) and ~12 (EEMS; 109.7 kg h$^{-1}$)
higher cumulatively for all sampled facilities. However, EEMS emission data for 2019 agree slightly better with the
observations taken in 2019: Top-down estimates are a factor of ~11 higher compared to the EEMS reported data (125.8 kg h$^{-1}$). Most $CH_4$ emissions of sampled installations and reported by EEMS are attributed to venting (35% - 96%) besides emissions
arising from the operation of turbines and engines (0.1% - 50%). It is worth noting that for all platforms listed in EEMS, zero
flaring emissions are reported. During the flights no visible flaring was observed. Nevertheless, flaring is stated to have a share
of 3% of the Southern North Sea region's total $CH_4$ emissions in 2019 (OGA, 2020). The Global Gas Flare Catalog 2019 from
the Earth Observation Group at the Payne Institute for Public Policy (Elvidge et al., 2015; Elvidge et al., 2013), which uses
VIIRS data, shows flaring in the North Sea region. However, for the sampled installations no flaring is observed in 2019,
which confirms the inventories zero flaring claim at least for the sampled installations.

As discussed in section 2, EEMS data is fed into the NAEI inventory, hence an agreement of both inventories for 2018 should
be expected. A comparison between NAEI data and EEMS data from 2018, however, shows that the numbers are consistent
only for two (P1, P2) out of five UK platforms. The NAEI 2018 value is smaller for P3 and higher for P4, P5 and P6 compared
to EEMS 2018. NAEI inventory emissions are expressed as a sum of all platforms associated with a field. In contrast, EEMS
emissions are listed for one specific platform, also in the case of multi-platform complexes (P4, P5). Those platforms might
be interpreted as being representative platforms with the reported emissions being aggregated emissions for the complex.
However, since EEMS informs NAEI, it is not clear why both inventories differ for some of the installations in this study.

Regarding P4, we used the FLEXPART (FLEXible PARTicle) dispersion model (Pisso et al., 2019) to attribute the measured
emission plumes to individual platforms located within the complex (see Appendix C). The platforms that the observed fluxes
were attributed to do not match with the (representative) platforms listed in EEMS 2018/2019.

The discrepancy to UK national inventories detected in this study is higher than reported in previous airborne studies of other
offshore regions. Zavala-Araiza et al. (2021) estimated offshore $CH_4$ emissions in the Sureste Basin, Mexico, to be more than
an order of magnitude lower than the values given in the Mexican greenhouse gas emission inventory. Gorchov Negron et al.
(2020) generated an airborne measurement-based inventory comprising offshore facilities located in the U.S. Gulf of Mexico.
They showed that for shallow-water facilities $CH_4$ emissions are more than a factor of two higher than the estimate of the U.S.
Environmental Protection Agency Greenhouse Gas Inventory (EPA GHGI) and the Gulfwide Offshore Activity Data System
(GOADS) inventory.





### 3.1.3 Comparison to facility-level reporting by platform operators for the survey date

As expected, the smallest discrepancy between top-down and bottom-up estimates exists for the comparison with emission data of individual facilities provided by platform operators for the specific survey day. Operator-based reporting was only

available for the five sampled Dutch installations (P7-P11). The facility-level estimates deviate by up to a factor of ~12 compared to the reporting, whereby two out of five facilities (P7, P10) are overestimated and another two facilities underestimated (P8, P11). According to additional information on emission types provided by the operators, $CH_4$ emissions arise from venting and fugitives for 4 out of 5 installations (P7-P10). For P11 no venting or flaring was recorded, although $CH_4$ was detected during the measurements conducted downstream. The measured emissions might be attributed to fugitives, which are not excluded by the operator in this case. Flaring emissions are explicitly excluded only for two out of five

installations (P10, P11). For P7-P9 flaring emissions could contribute, though. P9 is reported as offline on the survey day, which agrees with the measurements showing no elevated $CH_4$. For all sampled Dutch installations together, we find that our estimated flux of $(122.7 \pm 9.7)$ kg h$^{-1}$ deviates by a factor 0.7 (ranging from 0.35-12 for individual facilities) from reported values (183.1 kg h$^{-1}$). A comparison with operator-reported data for offshore installations in the Norwegian Sea by Foulds et

al. (2022) shows that although there are deviations for individual facilities, reported data agree similarly well on aggregate for a larger sample size (18 facilities) with the measured fluxes being smaller than reported emissions by a factor 0.8 (ranging from 0.1-22 for individual facilities).

### 3.1.4 Comparison to a ship-based top-down study

The planning for the flight on 2019/05/06 around Dutch installations relied on a ship-based top-down study conducted by the

Netherlands Organisation for Applied Scientific Research (TNO) in 2018 (Hensen et al., 2019). With the aim to derive $CH_4$ emission fluxes, measurements were taken at distances up to ~3 km downwind of 33 platforms in November 2018. $CH_4$ was measured with a QCL-ILC spectrometer (Aerodyne Research, Inc.) and a Picarro instrument, whereby the inlet was installed at 35 m above sea level. The results shown in Table 1 were obtained by combining the measurements with a Gaussian plume model and a tracer-release experiment. The derived fluxes range from 10 kg h$^{-1}$ to 252 kg h$^{-1}$. For P8 and P11 our fluxes are

within the range of the determined fluxes from the ship-based study, whereas in case of P7 and P10 our measured fluxes are smaller. For the studied 4 Dutch facilities in aggregate, our measured fluxes $((122.7 \pm 9.7)$ kg h$^{-1})$ are smaller with respect to the ship-based measurements (216.7 kg h$^{-1}$ - 536.8 kg h$^{-1}$) and deviate by factor 0.2-0.6.

### 3.2 Correlation between $CH_4$ and $C_2H_6$ for all platforms

For all sampled installations for which enhanced $CH_4$ was detected, we observe clear correlations with co-emitted $C_2H_6$, which

is an indicator for fossil fuel emissions (Lowry et al., 2020; Peischl et al., 2018; Hausmann et al., 2016; Smith et al., 2015). $C_2H_6$ to $CH_4$ molar ratios of fossil fuels depend on the type of field/reservoir (gas, gas condensate, oil). Since the Southern North Sea region contains predominantly dry gas fields with relatively low gas condensate (wet gas) production, we expect





low $C_2H_6$ to $CH_4$ ratios ranging from 1-5% (dry gas) and 5-10% (gas condensate) (Xiao et al., 2008, Jones et al., 1999) or 1-6% (dry gas) and > 6% (wet gas) (Yacovitch et al., 2014, Whiticar et al., 1994). We calculate an $C_2H_6$ to $CH_4$ molar ratio for

each transect from the integrated plume area of the respective $CH_4$ and $C_2H_6$ enhancement, and take the average over all transects for each sampled installation. Measured values range from 2.5% to 7.8%. We compare the measured ratios to reported values from the OGA Shell/ExxonMobil Geochemistry Database for Central North Sea (2017) for UK sites and the NLOG for Dutch sites (for all measured and reported values see Table D1 in the Appendix D). The reported $C_2H_6$ to $CH_4$ ratio for the UK installation P4 is 0.3% higher than the observed value. Compared to ratios measured downstream of Dutch facilities, the

reporting underestimates the measurements, except for one facility (P11). In general, the dry gas and gas condensate binary categorization matches for the observed and reported ratios.

**3.3 Correlation between $CH_4$ and $CO_2$ for selected platforms**

Enhanced $CO_2$ mole fractions accompanied the $CH_4$ enhancements at five installations (P1, P2, P4, P7, P10) indicating a combustion source from either flared $CH_4$ or other combustion sources such as turbines or engines. For P8 and P11 $C_2H_6$ was

enhanced while no $CO_2$ enhancement was observed (< LoD). Figure 3 shows the time series for a transect flown downwind of P1 with simultaneous enhancements in $CH_4$, $CO_2$ and $C_2H_6$ mole fractions as an example of the observed plumes. The $CO_2$ flux is determined from the gradient of a linear regression between the $CO_2$ and $CH_4$ enhancements since both species are detected by the same instrument (Picarro Analyzer). For three of the platforms (P1, P4, P10), $CH_4$ and $CO_2$ were well-correlated and $CO_2$ fluxes have been determined.

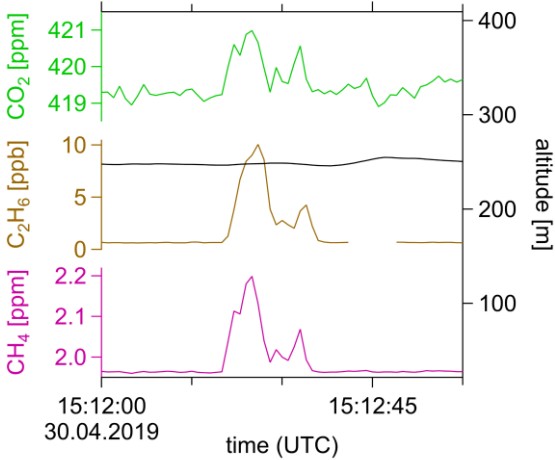

**Figure 3. Time series (1 Hz) of a transect at 250 m altitude downwind of P1: Coinciding elevations in $CO_2$ (green), $C_2H_6$ (brown) and $CH_4$ (magenta) molar ratios. $C_2H_6$ is a tracer for fossil fuel emission and $CO_2$ indicates a combustion source.**




Table D1 in the Appendix D shows the measured $CO_2$ fluxes along with inventory emission data from the UK point-source inventories NAEI and EEMS and Dutch operator data. For P1 the NAEI inventory matches within the uncertainties, but states lower emissions from the P4 platform complex. According to EEMS, which categorizes emissions into turbines/engines, fugitives, venting and flaring, $CO_2$ emissions arise mainly from the combustion of diesel and gas in turbines and engines. Only
for the platform complex P5 minor emissions from fugitives and venting are listed. In EEMS flaring emissions are zero for all UK platforms. This is inconsistent with data from the UK Oil and Gas Authority, which reports that 4% of $CO_2$ emissions in the SNS region are supposed to arise from flaring in 2019 (OGA, 2020). From the amount of $CO_2$ and $CH_4$ flaring emissions in 2019 in the SNS and Irish Sea region given in the Flaring and Venting Report (OGA, 2020), the unburnt fraction, i.e. the ratio of unburnt $CH_4$ to $CO_2$ from flaring emissions, is 6.4%. If we calculate this ratio for the sampled $CH_4$ and $CO_2$ plumes at
the UK platforms, we get higher ratios: 37.2% (P1) and 15.9% (P4). This means that either there is no flaring on the platform, or if some flaring occurred, there were additional $CH_4$ fugitive or venting sources. Comparing to Dutch operator data, we find that around two Dutch platforms (P8, P11) no simultaneously emitted $CO_2$ was detected, although Dutch operator data states $CO_2$ emission on the survey date. For P10 we derive a $CO_2$ flux half the size of the emissions reported for the survey date. Dutch operator data explicitly excludes flaring sources for P10 and P11 (see Table 1) and lists only combustion sources such
as turbines and engines. To sum up, from the measured total emissions we cannot clearly differentiate flaring from other combustion sources. But if there were any flaring sources, there must have been additional fugitive/venting $CH_4$ sources according to the measured $CH_4$ to $CO_2$ ratios.

**3.4 Loss rates**

In this section we determine loss rates, i.e. the ratio of gas lost to the atmosphere to dry gas production rates. We calculate the
amount of gas lost to the atmosphere from the determined $CH_4$ emission rates and the $CH_4$ mass % from the OGA Shell/ExxonMobil Geochemistry Database for Central North Sea (2017) for UK sites and from the operator data for Dutch sites. UK production rates are given as monthly values by OGA. We include production from upstream fields with only subsea wells and no platform infrastructure. Dutch production data was provided by Dutch operators for the specific survey day. For three UK facilities (P3, P5, P6) no emissions were detected, although they were producing during the month of survey.
According to the Dutch operator, P9 did not produce on the survey day, and we did not detect a plume either.

Determined loss rates for Dutch and UK sites are smaller than 1.0%, except for P4, which shows an higher loss rate of 2.9% (see Appendix E for individual production rates and loss rates). Besides the fact, that P4 is a multi-platform complex and relatively old, i.e. producing since 50 years, there is no indication of abnormal activities on the survey date.





### 3.4.1 Comparison with airborne studies in other regions (Norwegian Sea, Northern Gulf of Mexico)

Figure 4 depicts the determined $CH_4$ emission rates and production rates from this study compared to the results obtained in two other airborne studies conducted by Foulds et al. (2022) in the Norwegian Sea and Gorchov Negron et al. (2020) in the Northern Gulf of Mexico.

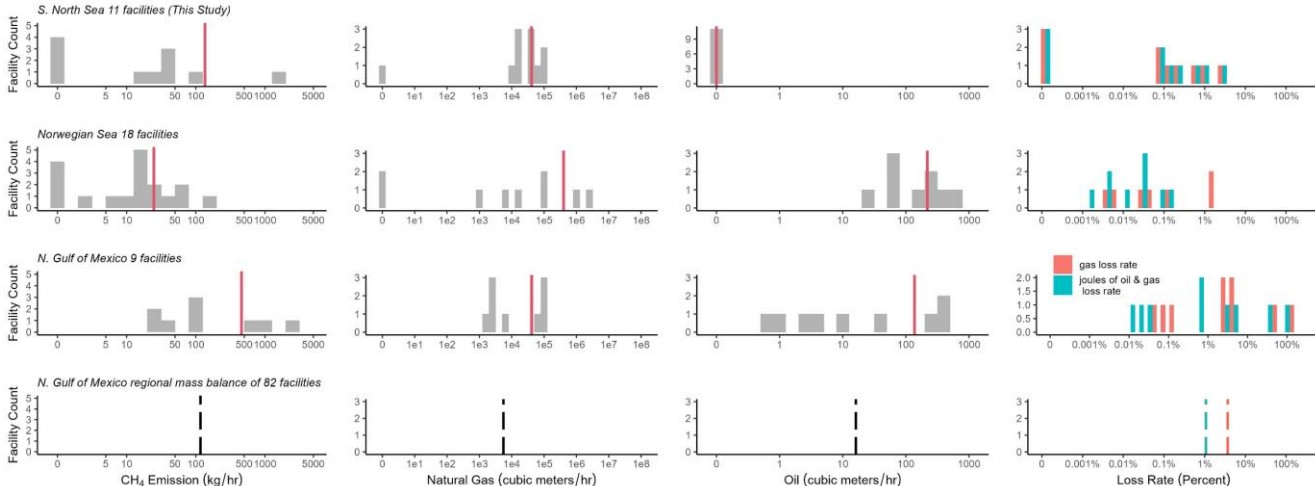

**Figure 4. Comparison of measured $CH_4$ emission rates (first column), corresponding natural gas (second column) and oil (third column) production rates and loss rates (fourth column) in the Southern North Sea (this study) with two other airborne studies conducted in the Norwegian Sea (Foulds et al., 2022)) and in the Northern Gulf of Mexico (Gorchov Negron et al., 2020). Red lines denote the respective average values. The dotted lines show the average value obtained in a regional mass balance in the Northern Gulf of Mexico. The facility count does not include satellite structures.**

The $CH_4$ emission fluxes for individual facilities, i.e. rates, calculated in this study compare to the emission rates determined 365 in the Norwegian Sea and in the Northern Gulf of Mexico (see left side of Figure 4). The emission rate of P4 is as high as the emissions measured around similar infrastructure types in the Northern Gulf of Mexico, i.e. multi-platform complexes in shallow water, which equally show emission rates higher than 500 kg h$^{-1}$. In the Gulf inconstant temporal variability of those infrastructure types was seen, what might correspond to the non-detectable emissions of the multi-platform complex P5. Comparing average absolute emission rates per facility (red vertical lines), the lowest average emission rates were determined 370 around 18 facilities in the Norwegian Sea (24 kg h$^{-1}$) and highest emission rates around 9 facilities in the Gulf (457 kg h$^{-1}$) with a factor of 19 difference. Our average emission estimate in the Southern North Sea is 136 kg h$^{-1}$ and compares well with the average absolute emission rate in a regional mass balance in the Gulf with a larger sample size (117 kg h$^{-1}$). When excluding the multi-platform complex P4, the Southern North Sea average emission estimate amount to 23 kg h$^{-1}$, which compares well with the average emission rate in the Norwegian Sea, where no multi-platform complex was sampled.

In contrast to the Southern North Sea, where gas (with little gas condensate) production dominates, in the Northern Gulf of Mexico natural gas is produced as a side product from oil exploitation (associated gas) and in the Norwegian Sea both oil and



gas production takes place. The natural gas production rates for the facilities in the Southern North Sea shown in the second column in Figure 4, are on average one order of magnitude smaller than in the Norwegian Sea and one order of magnitude higher than in the Gulf regional estimate, but almost the same value as the Gulf facility-wise estimate. Average oil production rates in the Norwegian Sea and in the Northern Gulf of Mexico are comparable.

Total loss rates, i.e. all gas lost to the atmosphere divided by total production rates in the respective region, can be determined either from gas production only or from the sum of oil and gas production. Thereby, we convert oil and gas production rate units according to the energy content. Considering only gas production, the total loss rate in the Southern North Sea (0.50% (0% - 2.8%)) is one order of magnitude higher than in the Norwegian Sea (0.02% (0.003% - 1.6%)) and one order of magnitude smaller than in the Gulf. The latter amount to 1.9% (0.04% - 128%) for the facility-level measurements and 3.7% for the regional measurements. Including oil production, total loss rates in the Norwegian Sea (0.01% (0.001% - 0.2%)) and in the Gulf (0.51% (0.01% - 112%) for the facility study; 1.1% for the regional study) are reduced. Thus, total loss rates in the Southern North Sea and in the Gulf compare to each other, when including oil production, and total loss rates in the Norwegian Sea are still one order of magnitude smaller compared to the other regions, but span over 3 orders of magnitude.

## 4 Conclusion

We report $CH_4$ flux estimates for six UK and five Dutch offshore gas production installations in the Southern North Sea derived from airborne measurements conducted in spring 2019. We identified the observed $CH_4$ enhancements as emissions arising from natural gas based on co-emitted $C_2H_6$ and derive $C_2H_6$ to $CH_4$ ratios for each offshore installation. Comparison with a ship-based top-down study conducted around Dutch facilities in 2018 (Hensen et al., 2019) shows that our derived $CH_4$ fluxes deviate by a factor 0.2-0.6 being smaller with respect to fluxes derived by Hensen et al. Our $CH_4$ flux estimates were compared with different bottom-up inventories available for this region, including the Global Fuel Exploitation Inventory (GFEI) (Scarpelli et al., 2019), the UK Environmental and Emissions Monitoring System database (EEMS), the UK National Atmospheric Emissions Inventory (NAEI), and direct facility-level reporting by Dutch operators. In general, the comparison for individual facilities shows a large discrepancy between the top-down derived emissions and all bottom-up (inventory and reported) estimates, which may be expected because of the nature of single snap-shot measurements per facility in this study and potential temporal variability per facility demonstrated via repeat measurements by Foulds et al. (2022). The largest discrepancy exists with the annual emission data from the globally gridded GFEI inventory for the year 2019, showing that measured aggregated emissions from UK and Dutch sites are higher by a factor of ~ 21 and ~ 279, respectively. On the one hand, these high discrepancy factors reflect the weaknesses in using global inventories based on Tier 1 methods for field-specific emissions characterizations, especially when comparing with snap-shot measurements. On the other hand, Dutch UNFCCC reported emissions, which the inventory is based on, are much smaller compared with UK UNFCCC reporting and could give rise to the exceptionally large factor for Dutch sites. Our top-down emission fluxes for all sampled UK installations in aggregate deviate from UK national annualized emission data from NAEI and EEMS for the year 2018 by factors of 6 and

12, respectively. Surprisingly, NAEI and EEMS inventory data are mutually inconsistent for four out of six installations,
although the NAEI is based on the EEMS operator-based reporting. Latest UK national inventory data available for 2019 from
EEMS deviate slightly less from the measurements with the latter being a factor 11 higher for all sampled UK facilities in
aggregate. According to the EEMS inventory, $CO_2$ emissions measured around UK facilities and correlating with $CH_4$
emissions are solely attributable to combustion sources (turbines, engines) while flaring emissions are reported as zero for both
$CO_2$ and $CH_4$. The measurements in this study cannot differentiate flaring from other combustive sources, and thus rule out
flaring. Still, the measured ratios of emitted $CH_4$ to $CO_2$ point at existing venting/fugitive $CH_4$ sources, whereby flaring sources
could be contributing.

As expected, the best agreement with our flux estimates exists with facility-level reporting from Dutch operators for the specific
survey date. The measurements deviate by a factor of 0.7 (0.35-12) and are smaller with respect to Dutch reported emissions
for all sampled facilities in aggregate. Our results for operator-based facility-level reporting compare very well to a study
conducted in the Norwegian Sea by Foulds et al. (2022), which find their measurements deviating by a factor 0.8 and being
smaller compared to the reporting by operators. We conclude that for sites with operator-based facility-level reporting in Dutch
waters, – as suggested in the reporting framework Oil and Gas Methane Partnership 2.0 (www.ogmpartnership.com) – the
highest accuracy is demonstrated compared to measurements. The adoption of facility-level estimation in national inventories
would be expected to increase the accuracy of national $CH_4$ emissions accounting for the offshore oil and gas sector. To
improve comparisons of top-down and bottom-up observation and resolve discrepancies, generating bottom-up inventories at
facility-scale and accounting for temporal variability when including top-down measurements would be extremely valuable.

A regional comparison to airborne studies in the Norwegian Sea (Foulds et al., 2022) and in the Northern Gulf of Mexico
(Gorchov Negron et al., 2020) shows that the absolute facility-level emission rates agree with the general distribution found
in other offshore basins. This is despite differing gas production rates, which span two orders of magnitudes across geographies.
Including oil production rates, total loss rates of the Southern North Sea compare to total loss rates in the Gulf, whereas loss
rates in the Norwegian Sea are one order of magnitude smaller. As a consequence of the similar absolute emission rates,
mitigation is needed virtually equally across geographies. Further, average absolute emission rates in this study are
substantially larger in the UK compared to NL, which is largely driven by one super-emitter in the UK. The emission of the
super-emitter is as high as the emissions measured around similar infrastructure types (multi-platform complexes in shallow
water) in the study in the Northern Gulf of Mexico, but additional sampling in future studies is needed to investigate
representativeness.



# Appendices

## A. Example for flux calculation for P1

In the following, the CH$_4$ flux calculation is illustrated by using observations of platform P1 on 30 April 2019.
Measurements were performed downwind at a distance of around 2-3 km from the platform (wind direction (179.5 $\pm$ 29.8) °; perpendicular wind speed $V_{\perp}$ = (3.2 $\pm$ 1.5) m/s). To fully capture the emitted CH$_4$ plume dispersed within the boundary layer, which extended up to (420 $\pm$ 20) m, vertically stacked transects were flown between 97 m and 305 m. Figure A.1 shows the downwind horizontal transects with CH$_4$ mole fractions color-coded and the corresponding time series. CH$_4$ enhancements were detected in all seven transects. We calculated CH$_4$ fluxes for each transect resulting in a total flux of (86.4 $\pm$ 18.9) kg h$^-$
$^1$. The uncertainty is given for confidence intervals of 1 standard deviation and arises mainly due to wind measurements.

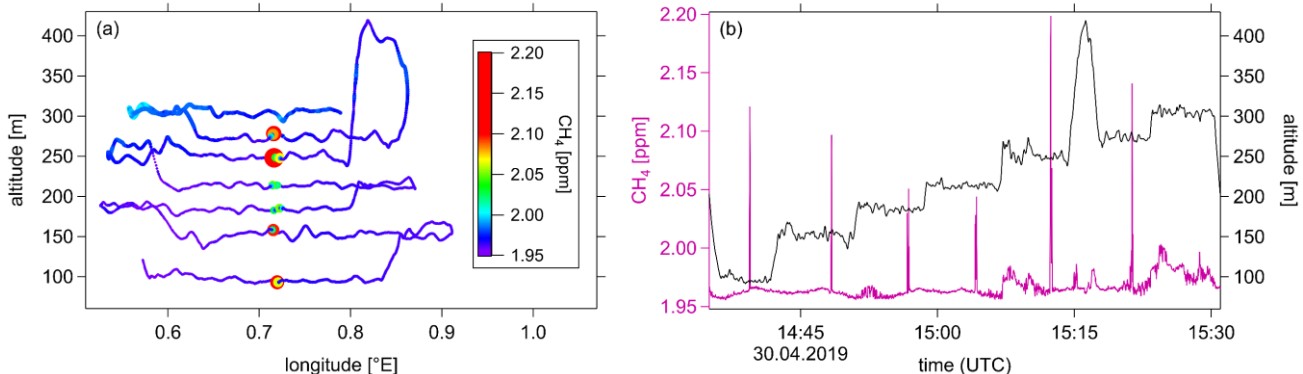

**Figure A1. Example for measurements downwind of platform P1 during the offshore flight on 30 April 2019: (a) horizontal transects at altitudes between 94 m and 304 m above sea level. CH$_4$ enhancements are elucidated with a color-scale, whereby the size of plotted symbols is scaled to CH$_4$ molar ratios. (b) corresponding CH$_4$ time series.**

## B. Uncertainty analysis for flux calculation

We use the Gaussian error propagation to determine the uncertainty of the flux calculation, represented as confidence intervals of 1 standard deviation (see eq. (1) and (2) in subsection 2.2). The uncertainties of the calculated CH$_4$ fluxes for each layer $i$
result from the uncertainties of each measured parameter $q$ (eq. B1). These parameters are the elevated CH$_4$ molar ratios $\Delta C_i$, wind speed $V_{\perp}$, pressure $p_i$, temperature $T_i$, plume width $\Delta x_i$ and plume height $D_i$. We calculate the uncertainty $u$ of the total flux with equation B2.

$$u(Flux_i) = \overline{Flux_i} \cdot \sqrt{\sum_q^{parameters} \left(\frac{u(q)}{\bar{q}}\right)^2} \tag{B1}$$

$$u(Flux_{total}) = \sqrt{\sum_i^{transects} u(Flux_i)^2} \tag{B2}$$

For $\Delta C_i$ the CH$_4$ molar ratios measured in- ($C_i$) and outside ($C_0$) of the plume are used (eq. (A3)). Both $C_i$ and $C_0$ have a systematic uncertainty resulting from the Picarro instrument uncertainty of 1.2 ppb (France et al., 2021). The background molar




ratio at each point $j$ within the plume boundaries $a$ and $b$ is determined from an interpolation between $C_{0,a}$ and $C_{0,b}$, which are the mean $CH_4$ molar ratios within 30 s before and after the plume. The uncertainty of the interpolated background at each point $u(\Delta C_{0,j})$ is calculated from the standard deviations $\sigma_{0,a}$ and $\sigma_{0,b}$ of $C_{0,a}$ and $C_{0,b}$ (eq. (A4)). The parameter $n$ denotes the number of points within the plume boundaries.

$$u(\Delta C_i) = \sqrt{\sum_a^b (u(C_{i,j})^2 + u(C_{0,j})^2)} \tag{B3}$$

$$u(\Delta C_{0,j}) = \sqrt{(\sigma_{0,a} \cdot \frac{n_i-j}{n_i})^2 + (\sigma_{0,b} \cdot \frac{j}{n_i})^2} \tag{B4}$$

We determine the perpendicular wind speed from the average aircraft heading, measured average horizontal wind speed and average wind angle over all transects. The uncertainty of the perpendicular wind speed $u(V_\perp)$ is a result of the standard deviations and valid for all transects:

$$u(V_\perp) = \sqrt{\left(\frac{\partial V_\perp}{\partial \, heading} \cdot \sigma_{heading}\right)^2 + \left(\frac{\partial V_\perp}{\partial \, wind \, speed} \cdot \sigma_{wind \, speed}\right)^2 + \left(\frac{\partial V_\perp}{\partial \, wind \, angle} \cdot \sigma_{wind \, angle}\right)^2} \tag{B5}$$

For the uncertainties of pressure $u(p_i)$ and temperature $u(T_i)$ the standard deviations of the mean values across the plume and the 30 s background are taken.

The plume width is determined by the distance the aircraft covered while crossing the plume. Thereby, the velocity of the aircraft is multiplied with the time span of the plume. The uncertainty of the plume width $u(x_i)$ is derived from the uncertainty (standard deviation) of the measured velocity of the aircraft.

Since we assume a well-mixed plume within the boundary layer, the uncertainty of plume height $u(D_i)$ is characterized by the uncertainty arising from the estimation of the boundary layer height. Therefore, $u(D_i)$ is only relevant for the uncertainty of the flux calculated for the uppermost layer.

## C. FLEXPART Dispersion Model: Example footprint analysis for the multi-platform complex P4 (backward simulation)

The model study concludes, that 9 out of 19 platforms of the complex could have contributed to the measured $CH_4$ enhancement (flight track with color-coded $CH_4$ in Figure C1). None of the possible emitters is listed in the inventories as single platforms.





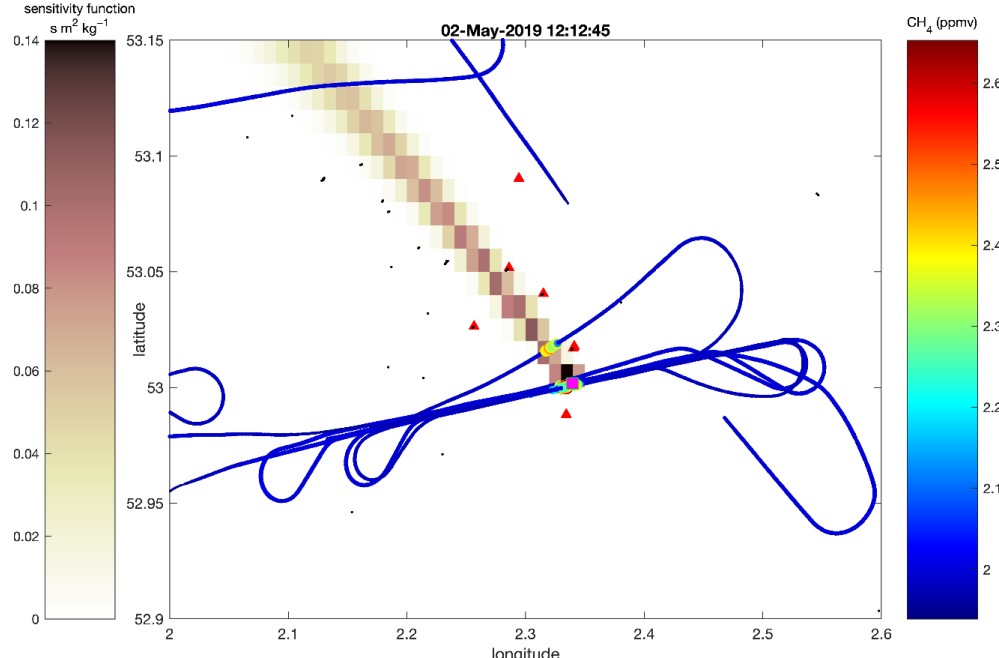

**Figure C1: Footprint analysis for the multi-platform complex P4 (backward simulation) sampled during flight 327: The flight track is shown with color-coded CH₄ in units of ppm. The particle density is shown with a brownish scale. Black markers symbolize the installations in the area and red markers show all installations which could have contributed to the measured plume.**


**D. Comparison of C₂H₆ to CH₄ ratios and CO₂ fluxes with reported values**



**Table D1. Measured and reported (OGA, NLOG) $C_2H_6$ to $CH_4$ ratios for all sampled platforms. All installations for which $CH_4$ was enhanced were accompanied by co-emitted $C_2H_6$. No detected $C_2H_6$ and $CO_2$ enhancements are indicated with $< LoD$. Measured $CO_2$ emissions are compared to UK inventory data (NAEI, EEMS) and Dutch operator data. Information on emission processes is given for venting (vent), fugitives (fug) and flaring (flar). $CO_2$ emissions are rounded to two significant digits.**

| Survey date | | 30 April 2019 | | | 2 May 2019 | | | 6 May 2019 | | | | |
|---|---|---|---|---|---|---|---|---|---|---|---|---|
| Country | | UK | | | UK | | | NL | | | | |
| Installation | | P1 | P2 | P3 | P4 | P5 | P6 | P7 | P8 | P9 | P10 | P11 |
| $C_2H_6/CH_4$ [%] | This study | 4.3 ± 0.1 | 2.5 ± 0.6 | <LoD | 2.9 ± 0.2 | <LoD | <LoD | 5.6 ± 0.2 | 4.3 ± 0.4 | <LoD | 7.8 ± 0.4 | 5.3 ± 0.2 |
| | OGA/NLOG | n.a. | n.a. | 3.3 | 3.2 | 4.9 | 4.3 | 3.3 | 3.6 | 4.3 | 6.0 | 6.1 |
| $CO_2$ flux [kg/h] | This study | 640 ± 230 | weak correlation | <LoD | 21770 ± 5230 | <LoD | <LoD | weak correlation | <LoD | <LoD | 5300 ± 680 | <LoD |
| | NAEI (2018) | 410 | 390 | 200 | 3110 | 4160 | 1070 | n.a. | | | | |
| | EEMS (2018/2019) all | 1490 / 1490 | 1430 / 1210 | 740 / 740 | 11400 / 17110 | 15260 / 11880 | 3930 / 2180 | n.a. | | | | |
| | turbines, engines | 1490 / 1490 | 1430 / 1210 | 740 / 740 | 11400 / 17110 | 15260 / 11880 | 3930 / 2180 | | | | | |
| | fug | 0 / 0 | 0 / 0 | n.a. / n.a. | 0 / 0 | 0.07 / 0.05 | 0 / 0 | | | | | |
| | vent | 0 / 0 | 0 / 0 | 0 / 0 | 0 / 0 | 0.07 / 0.08 | 0.02 / 0.02 | | | | | |
| | flar | 0 / 0 | 0 / 0 | 0 / 0 | 0 / 0 | 0 / 0 | 0 / 0 | | | | | |
| Reporting by operators | | n.a. | | | | | | 16520 turbines, furnaces, vent | 330 turbines, vent | 0 | 10690 turbines, power generators | 570 Turbines, engines, furnaces, diesel |


**E. Production rates and loss rates (including non-emitting installations)**

Figure E1 shows gas lost to the atmosphere, which is calculated from $CH_4$ emission rates and the $CH_4$ mass % (UK sites: OGA Shell/ExxonMobil Geochemistry Database for Central North Sea (2017); NL sites: operator data). The determined loss rates are the ratio of gas loss and dry gas production, i. e. normalized $CH_4$ emissions against natural gas production rates.

Table E1 shows platform production rates along with calculated loss rates. No loss rates were determined for installations, where emissions were below detection limit and thus, no enhancements measured (abbreviation "no enh."). Z1-Z8 are non-

emitting installations from fly-bys. Individual platform production data for 2019 were taken from the UK Oil and Gas Authority (OGA), the NLOG and operator reported data. UK production rates are given as monthly values by OGA. Thereby, we include production from upstream fields with only subsea wells and no platform infrastructure. Dutch production data was provided by Dutch operators for the specific survey day.

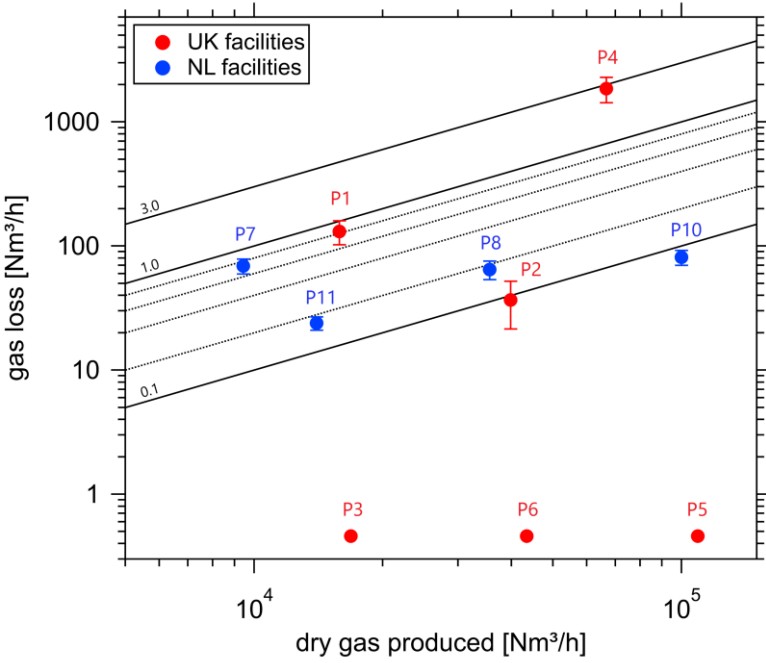

**Figure E1. Gas lost to the atmosphere against the amount of dry gas produced in norm cubic meters (Nm³) per hour (UK: OGA, NL: operator data). Dutch platforms are shown in blue, UK platforms in red. Note that no downwind enhancements were detected for 4 installations (P3, P5, P6, P9) with only P9 (NL installation, excluded) not producing. Lines of constant loss rates (%) are shown in black.**





**Table E1. Reported production rates and calculated loss rates for sampled UK (P1-P6) and Dutch (P1-P11) installations. Z1-Z8 are (non-emitting) installations from fly-bys.**

| facility | Dry gas production (OGA (UK), NLOG (Dutch)) [Nm³/month] | Operator reported gas production [Nm³/day] | loss rate [%] | loss rate uncertainty [%] | Start of production [yr] |
|---|---|---|---|---|---|
| P1 [a] | 10238885 (+ 98 Nm³ gas condensate) | n.a. | 0.83 | 0.19 | 1988 |
| P2 [b] | 25765475 | n.a. | 0.09 | 0.04 | 1990 |
| P3 | 28090814 (+ 44 Nm³ gas condensate) | n.a. | no enh. | no enh. | 1967 |
| P4 [b] | 44571997 (+ 194 Nm³ gas condensate) | n.a. | 2.79 | 0.68 | 1968 |
| P5 | 72934875 (+ 72 Nm³ gas condensate) | n.a. | no enh. | no enh. | 1968 |
| P6 [b] | 11259835 (+ 150 Nm³ gas condensate) | n.a. | no enh. | no enh. | 1969 |
| P7 | 855993 | 226383 | 0.73 | 0.11 | 1977 |
| P8 | 11049455 | 854000 | 0.18 | 0.03 | 1983 |
| P9 | 0 | 0 | no enh. | no enh. | 1991 |
| P10 | 28340954 | 2400000 [c] | 0.08 | 0.01 | 1994 |
| P11 | 13314491 | 335996 | 0.17 | 0.02 | 2005 |
| Z1 [d] | 3145322 (+ 3 Nm³ gas condensate) | n.a. | no enh. | no enh. | 1993 |
| Z2 [d] | 14321737 (+ 198 Nm³ gas condensate) | n.a. | no enh. | no enh. | 2003 |
| Z3 | 0 | n.a. | no enh. | no enh. | 1987 |
| Z4 | 3794100 | n.a. | no enh. | no enh. | 1985 |
| Z5 | 0 | n.a. | no enh. | no enh. | 2007 |
| Z6 | 0 | n.a. | no enh. | no enh. | 2004 |
| Z7 | 13542079 | n.a. | no enh. | no enh. | 2002 |
| Z8 | 3251685 | n.a. | no enh. | no enh. | 1990 |

[a] Zero gas production for the month of survey. Production only of delivering subsea wells.

[b] including one delivering subsea well

[c] gas production with little gas condensate (gas condensate is injected back into export gas)

[d] unmanned installation


## Code and data availability

Access to the data is provided via request at the British Antarctic Survey Polar Data Centre.



**Author contributions**

The paper was written and figures were prepared by MP with contributions from AMGN. Modelling work was done by IP.

All authors contributed to the discussion. The experimental design and flight planning were performed by GA, JL, TLC and DL. Aircraft set-up and in-flight measurements were performed by PB, PD, SA, SY, AW, TLC and JF. Laboratory measurements were made by REF, and data processing and calibrations were performed by JF, LH, PB, JS, PD.

**Competing interests**

The authors declare that they have no conflict of interest.

**Acknowledgements**

This work was funded under the Climate & Clean Air Coalition (CCAC) Oil and Gas Methane Science Studies (MSS) programme, hosted by the United Nations Environment Programme. Funding was provided by the Environmental Defense Fund (EDF), Oil and Gas Climate Initiative, European Commission, and CCAC.

**Financial Support**

This research has been supported by the Climate & Clean Air Coalition (CCAC) (grant no. DTIE18-EN018).

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
