# Peer review of "Aircraft-based mass balance estimate of methane emissions from offshore gas facilities in the Southern North Sea"

_Atmospheric Chemistry and Physics, 2022_

## Author Comment (AC1)

We thank both reviewers for their constructive and insightful comments, which will help us to improve the manuscript. We address each reviewer comment below (in bold) and describe the revisions made to the paper where appropriate.

**Comment Reviewer 1:**

This paper has estimated methane emissions from offshore gas facilities in the Southern North Sea based on airborne observations of $CH_4$, $CO_2$, $C_2H_6$. A mass balance approach was applied to the observed $CH_4$ enhancements downwind of the targeted facilities to derive the emissions. Furthermore, the estimated emissions were compared to inventory studies, operator-based facility-level reporting, and other relevant and similar studies. Top-down quantification of $CH_4$ emissions from offshore oil and gas facilities is much needed to independently evaluate various $CH_4$ emissions, either expected or unexpected, and these efforts are very useful to help mitigate $CH_4$ emissions. The manuscript is well structured and well written, and will be suitable for publication after addressing my comments below.

Assuming a vertically well-mixed plume within the PBL and observing the enhancements that vary significantly across the transects at different heights are contradictory. It may be justifiable to assume the entrainment flux is small, and the top most transect can be extrapolated to the top of the PBL. Can the author justify the extrapolation to the sea surface? Note that the surface layer usually has distinct mixing scheme compared to the PBL. Also, LES simulations indicate that plumes downwind of point sources are not fully mixed kilometers from the source, and only time-averaged plumes show the Gaussian shape, e.g., Raznjevic et al., 2022.

**→ The vertical well-mixedness can be considered as an approximation to the hipothesis required by certain forms of the mass balance technique. Regarding the upper PBL, we assume the entrainment flux is small and extrapolate the top most transect to the top of the PBL. However, the key height is not that of the marine PBL but that of the mixed layer in which the plume resides (it is reasonable to assume they coincide, but it is conceivable that they may not be the same). We use the PBL height as the maximal upper plume boundary only where $CH_4$ enhancements were detected up to the highest transect of the aircraft (P1, P2, P3, P7). When no $CH_4$ enhancements were measured, the uppermost transect can be used to reduce uncertainties associated with the height of the PBL using such height as the top of the plume (P8, P10, P11).**

**In terms of mixing assumptions, our measurements took place at 2-7 km downwind distance of the target platforms. At these distances and for release heights of 35-100 m (average platform and venting stack heights) it is reasonable to expect the plume to be mixed to the sea surface (Raznjevic et al., 2022, has shown in LES simulations plumes mixing to the ground for these distances and release heights).**

**The minimum flux, i.e. from the lowermost to the uppermost layer and without interpolation to the sea surface and to the PBL top, lies within the new calculated uncertainty range of the fluxes (23-70%; see answer below). The description of actual turbulent motion during the campaign is not available and beyond the scope of the present work. The errors resulting from heterogeneities caused by turbulent motions can be considered estimating the corresponding associated**

uncertainties. In other words, where any spatial extrapolation/assumption has been used, the flux corresponding to the flux plane area of any spatial extrapolation is also added directly to the flux uncertainty to ensure that the uncertainty is conservative. This is why our uncertainties are typically a high percentage of the flux itself.

→ Indeed, for the shown example (Fig. A1), the enhancement strength varies with the strongest enhancements found above 200 m altitude ASL (layers 4-6 from 7 layers). This points to two methane sources with one being very likely a buoyant plume because of enhanced $CO_2$. Of course, the measurement method benefits from a large number of transects within the PBL to get a higher resolution of the plume in the vertical. We have introduced the vertical layering to calculate fluxes for the measured transects separately. In this way, we attempt to account for a non-uniform vertical plume structure.

The comparison of the mass balance estimates with operator-based reporting is highly appreciated. From the main text, I understand that operator-based reporting is provided daily. As maintenance activities and planned venting may vary over hours, I wonder the detailed info was provided by the operator as well. This may potentially explain the still quite large discrepancies between the mass balance estimates and the operator-based reporting.

→ The operator-based reported emissions are daily values with no higher time resolution (e.g. of hours). It is reported whether the platform was producing / online or offline. If it was offline, the period of time is given. This is the case for P9 (offline on the survey day) and P10 (offline during time of flight, while emissions were still measured). The reporting also includes information on the emission processes, i.e. whether emissions arise from venting, fugitives or flaring. However, the information is not complete for the set of sampled platforms as there is no information given on possible flaring activities for P7 and P8. This could give rise to the discrepancies. We have now added more details on time resolution of the operator-based reporting and the possible reasons for discrepancies to the revised manuscript.

Lines 192-195:

*"The reporting comprises information on the status of the installation (producing or offline on an hourly basis), the total amount of gas produced on the survey day and $CH_4$ and $CO_2$ emissions on the survey day including additional information on emission types and sources (venting, flaring, fugitives)."*

Lines 299-303:

*"P10 is reported as offline during the time of flight, while emissions are still measured and smaller than the reported venting $CH_4$ emissions for the survey day. For P11 no venting or flaring was recorded by the operator, although $CH_4$ was detected during the measurements conducted downstream. The measured emissions might be attributed to fugitives, which are not excluded by the operator in this case. Flaring emissions are explicitly excluded only for two out of five installations (P10, P11). For P7-P9 flaring emissions could contribute, though."*

The derivation of flux uncertainties is well described. However, some important info is lacking for a reader to understand why the uncertainties of the estimated fluxes are rather small compared to other studies, e.g., what are the wind speed and the wind speed and direction variabilities? Has the upwind transect been considered as backgrounds? How much differences are found in the calculated fluxes if a Kriging method is used to interpolate/extrapolate the plumes? What's the flux uncertainty caused by the uncertainty in the PBL height? How is the uncertainty of the sum emissions of all facilities derived from the individual uncertainties?

→ **The uncertainty of the wind measurement (especially direction) is the biggest contributor to the total uncertainty of the flux calculation (typically > 90% of the uncertainty). Measured wind speed and wind direction measurements used to calculate fluxes are average values from all transects and vary from 1-3 m/s (19-70% relative uncertainty at $1\sigma$) and 8-39° (2-19% relative uncertainty at $1\sigma$), respectively. The uncertainty of the perpendicular wind speed used for the flux calculation and calculated from the measured wind speed, wind direction and aircraft heading, lies between 1 and 3 m/s (22% and 70% relative uncertainty at $1\sigma$).**

**For the determination of the PBL height, which is based on the observation of a sudden sharp change in gradient of the measured PBL potential temperature profile, we assume an uncertainty of 20-32 m for the three measurement flights, which accounts for less than 10% for the flux calculated in the uppermost layer.**

**The uncertainty of the calculated flux of each transect is derived using the Gaussian error propagation method. Since the fluxes calculated for each transect are independent and therefore added up, the total uncertainty is the sum of the uncertainties of the fluxes calculated for each transect. The latter will be changed in the manuscript (before we used a Gaussian error propagation for the total error over all transects) and total uncertainties for the sampled platforms range from 23% to 70% (on average 39%).**

**Lines 228-229:**

***"The relative uncertainties of the determined fluxes range from 23% to 70% with the wind measurements as main contributor (> 90%)."***

**Lines 461-462 (Appendix B):**

***"The total uncertainty is the sum of the uncertainties of the fluxes calculated for each transect (eq. B2)."***

**Lines 485-489 (Appendix B):**

***"The uncertainty of the wind measurement is the biggest contributor to the total uncertainty of the flux calculation (typically 90%). Measured wind speed and wind direction measurements show variations ranging from 1-3 m/s and 8-39°, respectively. The uncertainty of the perpendicular wind speed $u(V_\perp)$ used for the flux calculation lies between 1-3 m/s (22-70% relative uncertainty at $1\sigma$). The uncertainty of plume height ranges from 20-32 m and accounts for less than 10% for the flux calculated for the uppermost layer."***

→ **We chose to use the background concentrations at the plume boundaries rather than the upwind concentrations, because either the upwind and downwind**

measurements show both a gradient in concentrations (e.g. increasing concentrations from west to east for P1 and P2) or because deviations of upwind and downwind background concentrations are small (below instrumental resolution).

→ Due to the rather sharp peaks, we decided for the interpolation mass balance method / layer method instead of Kriging, which is less able to represent highly-transient (spiky) data.

Some details:

Both Line 115 and Line 302 refer to the Aerodyne instrument using the same technique, please be consistent with the name of the instrument; also, please be consistent in using American and British spelling, e.g., analyzer, analyser, tuneable or tunable

→ Apologies. We will change the manuscript for a consistent spelling.

L130: should be mole fractions? As analyzers report concentrations in mole fractions, not molar ratios. Also for other occurrences throughout the text.

→ We agree. We will change this to mole fractions.

L131-132: not clear how backgrounds are calculated, please rephrase the sentence. It seems there is a typo here to interpolate between plume edge and side of the plume.

→ The background is calculated as the mean mole fraction measured over a 30 s time span before and after the observed plume. The beginning of the plume is defined as a measured concentration enhancement that is higher than 2 standard deviations of the background mole fractions. The background mole fractions during the time of flight through the plume are calculated using the average background mole fractions at either side of the plume and linearly interpolating in between to account for any  - typically extremely small - drift in background.

Lines 130-133 (and lines 465-472 in Appendix B):

"$\Delta C_i$ represents the difference of CH$_4$ mole fractions measured in- ($C_i$) and outside ($C_0$) of the plume ($\Delta C_i = C_i - C_0$). The background mole fractions $C_0$ during the time of flight through the plume are individually calculated for each transect. Thereby we use the average CH$_4$ mole fractions over a 30 s time span at either side of the plume and interpolate linearly in between to account for any drift in background"

L197-198: Can the authors clarify how the level of detection was derived? Not sure how the maximum uncertainty of all flux calculation parameters is defined?

→The level of detection was defined as the flux calculated from the maximum uncertainty of the measured parameters required for the flux calculation. The maximum uncertainty of the measured parameters (CH$_4$ molar ratios $\Delta C_i$, wind speed $V_\perp$, pressure $p_i$, temperature $T_i$, plume width $\Delta x_i$ and layer depth $D_i$)) are taken from the data of the three studied measurement flights.

Lines 203-205:

*"Under the prevailing conditions found during the three flights, the level of detection, which is a result of the maximum uncertainty of all measured flux calculation parameters (wind speed $V_\perp$, layer depth $D_i$, CH₄ enhancement $\Delta C_i$, pressure $p_i$, temperature $T_i$, plume width $\Delta x_i$), is 0.3 kg h$^{-1}$ (2σ)."*

L291: As P9 is reported offline, no $CO_2$ plume was detected, can the authors exclude flaring emissions from P9?

→ **The measurements downwind of P9 indicate no flaring emissions, since no CO₂ plume was detected, Further, the operator reports P9 as offline, which makes flaring emissions unlikely.**

L329-330, Figure 3, it would be nice to see the scatter plots of $CH_4$ vs. $CO_2$ and $CH_4$ vs. $C_2H_6$.

→**We agree. We will add the scatter plots in the appendix (Figure A2 in Appendix A).**

L455: eq.(A3) is found nowhere.

→ **Apologies. We have removed the reference.**

**Open comment:**

Hi there, I'm the Technical Director for the NAEI and have worked on the EEMS and UK inventory data for about 20 years. This is really useful research but it would be useful if possible to be able to view some of the underlying data per installation, to understand the findings more clearly. As regards the article, you state in the paragraph on line 265 (and then come back with a comment ("Surprisingly...") on line 409) that you would expect consistency between EEMS and the NAEI; this is not always the case. You have slightly mis-represented the NAEI, as the EEMS data is ONE OF SEVERAL data inputs to the NAEI emission estimates. We also use data such as EUETS data and also operator-reported activity data which is gathered by the North Sea Transition Authority via a mechanism called the Petroleum Producers Reporting System (PPRS). In the compilation of the NAEI estimates, we also conduct rigorous time series consistency and completeness checks on the raw EEMS data, and revise or gap-fill the EEMS data in using them for the national inventory. Therefore, it is not surprising to me that you are observing some inconsistencies between EEMS and NAEI. In general, I would expect EEMS to be a de-minimis value, so I am interested in the data for site P3; that disparity (EEMS>NAEI) could arise through data aggregation within the point source reporting process, but without seeing the data I can't comment further. One other very minor point is that there is a typo in the title of Table E1, which should read "...Dutch (P7 to P11) installations...". Kind regards, Glen Thistlethwaite.

→ **We describe the NAEI data based on the information from the UK Greenhouse Gas Inventory report (**[UK Greenhouse Gas Inventory, 1990 to 2021 (defra.gov.uk)](UK Greenhouse Gas Inventory, 1990 to 2021 (defra.gov.uk))**). We will change the manuscript based on the more detailed information on the NAEI inventory given in the open comment and from personal communication with Glen Thistlethwaite, which we are very thankful for. This information could indeed**

explain the differences we noticed between EEMS and NAEI, i.e. NAEI > EEMS, for all platforms except for P3, and we now discuss this in the paper.

**Lines 182-186:**

*"For offshore oil and gas installations it is based on the Emissions Trading Scheme (ETS) dataset for combustion and flaring sources and on the EEMS inventory for fugitives, venting and other sources such as oil loading (with combustion and flaring data only used if not available in ETS) (Brown et al., 2023; personal communication with the technical director for the NAEI). The inventory compilation process includes quality checks against other reporting systems such as the Petroleum Production Reporting System (PPRS), which also reports venting, flaring and gas use data."*

**Lines 280-284:**

*"As discussed in section 2, EEMS data is fed into the NAEI inventory, hence we expect that NAEI 2018 reported values are the same or higher than EEMS 2018 data. A comparison between NAEI data and EEMS data from 2018 shows that NAEI numbers are consistent with EEMS for two (P1, P2) and higher than EEMS data for three (P4, P5, P6) UK platforms. However, for P3 the NAEI reported value is smaller compared to EEMS 2018. This could either indicate an error in the EEMS reporting or it might be that the emissions of P3, which consists of 3 platforms, are misallocated in the NAEI."*

---

## Author Response (AR2)

We thank the reviewer for the second review, which will help us to improve the manuscript. We address comment below (in bold) and describe the revisions made to the paper where appropriate.

Thanks for the authors' detailed responses. A few minor comments:

1. The authors have done a good job of addressing my first comment, however, have not used the responses to improve the revised manuscript. It will be good if the authors clarify the assumption of the vertically well-mixed plume and at the same time use transects at multiple heights to further improve the calculation, and to estimate the uncertainties.

→ **We add more information of the approximation of the vertically well-mixed plume to the manuscript for clarification.**

**The mass balance method benefits of a large number of transects conducted downwind of an installation within the PBL. This is why the number of transects for the sampled installations in this study is minimum 4 and range up to 9 transects per installation. We use the measurements of all transects for the flux calculation of each sampled installation.**

**Lines 126-130:**

*"In general, the mass balance method is applied with the approximation that the plume is vertically well-mixed within the planetary boundary layer. However, to reduce the uncertainty of this approximation under the given meteorological conditions, we conduct horizontal transects at several altitudes to get a higher resolution of the dispersed plume in the vertical. Thereby, we subdivide the 2D vertical plane into discrete mixing layers to account for a possible non-uniformly spread plume."*

**Lines 144-147:**

*"We use all horizontal transects for the flux calculation with the highest transect, where enhancements are found, as the upper plume boundary. In the case where $CH_4$ enhancements were detected up to the highest transect of the aircraft, we use the boundary layer height as the maximal upper plume boundary assuming that the entrainment flux is small. "*

**Lines 205-207:**

*"The number of horizontal transects conducted downwind of the sampled installations and used for the flux calculation range from 4 to 9."*

2. Is the new calculated uncertainty range of the fluxes of 23-70% consistent with those shown in Figure 2? I also wonder whether the large uncertainties are associated with very low wind speed. As the wind speed of 1-3 m/s is relatively low.

**→ Yes, figure 2 and table 1 were updated with the new uncertainty ranges. The uncertainty is highest for P2 (70%) and lowest for P11 (23%). The wind speed variations/uncertainties range from 1-3 m/s, while actual wind speeds are moderate between 3-8 m/s (see line 205). In case of P2, there was a high uncertainty in wind speed relative to the low wind speed (3.0 $\pm$ 2.0) m/s, what is the reason for the total flux uncertainty of 70%.**

**Figure 2**

**Table 1**

**Line 205:**

***"The flight conditions during the flights selected for this study were generally good with moderate wind speeds (3-8 m/s)."***

**See Appendix B, Lines 498-508:**

***"The uncertainty of the wind measurement is the biggest contributor to the total uncertainty of the flux calculation (typically 90%). Uncertainties of wind speed and wind direction measurements range from 1-3 m/s (23-70% relative uncertainty at $1\sigma$) and 8-39° (2-19% relative uncertainty at $1\sigma$), respectively. The uncertainty of plume height ranges from 20-32 m and accounts for less than 10% of total uncertainty of the flux calculated for the uppermost layer."***

1. The authors indicated "The uncertainty of the wind measurement (especially direction) is the biggest contributor to …" From the calculated uncertainty values, the uncertainty of the wind speed is by far the largest. Not sure why the authors emphasized the wind direction.

**→ We apologize for this error in the first author response. We confirm that the uncertainty of the wind speed is the largest contributor.**

Can the authors consider adding a panel in Figure A2 to show the plume of C2H6 measured on Aerodyne and that of CH4 measured on Picarro? I feel that such a plot will be appreciated by readers.

**→ We add another panel in Figure A2 to show the enhancements in C2H6 and CH4 for peak 5 of P1, which is also shown in the first panel in Figure A2. Thereby, the areas under the peaks are highlighted, which are used to derive the C2H6 to CH4 (C2:C1) ratio.**

**Lines 334-335:**

***"As an example for the calculation, Figure A2 (b) in the Appendix A shows the simultaneous enhancements in $C_2H_6$ and $CH_4$ for peak 5 of P1."***

**Appendix A, Figure A2:**

[Figure]

**Figure A2.** (a) Scatter plot for co-emitted CO₂ downwind of platform P1. Enhanced CO₂ was found for two peaks at altitudes above 240 m.(b) Time series (1 Hz) of the transect at 250 m altitude downwind of P1 (peak 5): Coinciding elevations in C₂H₆ (brown) and CH₄ (magenta) mole fractions. The C₂H₆ to CH₄ (C2:C1) ratio is calculated from the fraction of the integrated peak areas (yellow) over the background mole fractions (gray) and over the time span of the peak (18 s).

**Appendix A, Lines 466-471:**

*"Panel (a) in Figure A.2 shows the scatter plot for CO₂ and CH₄ for platform P1, where enhanced CO₂ was found for two peaks at altitudes above 240 m. The observation of co-emitted CO₂ points to a buoyant plume adding up to the CH₄ plume at altitudes above 240 m. Panel (b) in Figure A.2. shows the time series of measured CH₄ and C₂H₆ for the transect at 250 m altitude downwind of P1 to illustrate the calculation of the C₂H₆ to CH₄ (C2:C1) ratio. The peak areas for C₂H₆ and CH₄ enhancements over the background are shown in yellow. The C2:C1 ratio is calculated by dividing the integrated peak area of C₂H₆ by the integrated peak area of CH₄, which results in a C2:C1 ratio of 4.3% in this case."*